# Solar-driven sugar production directly from $CO_2$ via a customizable electrocatalytic–biocatalytic flow system

Guangyu Liu[1,2,6], Yuan Zhong[1,6], Zehua Liu[1,6], Gang Wang[3,4], Feng Gao[1], Chao Zhang[1], Yujie Wang[1], Hongwei Zhang[1], Jun Ma[1], Yangguang Hu[1], Aobo Chen[1], Jiangyuan Pan[1], Yuanzeng Min[1], Zhiyong Tang[3,4,5], Chao Gao[1] ✉ & Yujie Xiong[1,2,5] ✉

Conventional food production is restricted by energy conversion efficiency of natural photosynthesis and demand for natural resources. Solar-driven artificial food synthesis from $CO_2$ provides an intriguing approach to overcome the limitations of natural photosynthesis while promoting carbon-neutral economy, however, it remains very challenging. Here, we report the design of a hybrid electrocatalytic–biocatalytic flow system, coupling photovoltaics-powered electrocatalysis ($CO_2$ to formate) with five-enzyme cascade platform (formate to sugar) engineered via genetic mutation and bioinformatics, which achieves conversion of $CO_2$ to $C_6$ sugar (L-sorbose) with a solar-to-food energy conversion efficiency of 3.5%, outperforming natural photosynthesis by over three-fold. This flow system can in principle be programmed by coupling with diverse enzymes toward production of multifarious food from $CO_2$. This work opens a promising avenue for artificial food synthesis from $CO_2$ under confined environments.

Crop plants cultivation, essentially based on natural photosynthesis that converts water and atmospheric $CO_2$ into carbohydrates, has been the primary way for food production for thousands of years[1]. Nevertheless, the food production is severely limited by the efficiency of natural photosynthesis, as the energy conversion efficiencies to biomass for most crop plants are only ~1% or less[2]. More crucially, traditional food production by cultivation is facing the challenges from sustainable development, such as overuse of chemical pesticides and fertilizers, as well as encountering geographical restrictions, such as global climate change, land scarcity, and shortage of fresh water[3]. As such, new approaches to enhancing the efficiency for food production,

while reducing the dependency on natural resources and avoiding environmentally harmful chemicals, are greatly desired to supplement the traditional food production.

From the viewpoint of sustainable development, solar energy should be utilized as the major energy input for efficient artificial synthesis of food directly from $CO_2$ via elaborate design and integration of systems, overcoming the limitations of natural photosynthesis in crop plants. If achieved, such an approach would not only offer a promising sustainable approach for food production, but also promote carbon-neutral economy. This beautiful blueprint encounters with a grand challenge. Electrocatalysis and photocatalysis, driven by

[1]Hefei National Research Center for Physical Sciences at the Microscale, Collaborative Innovative Center of Chemistry for Energy Materials (iChEM), School of Chemistry and Materials Science, University of Science and Technology of China, Hefei, Anhui 230026, China. [2]Suzhou Institute for Advanced Research, Nano Science and Technology Institute, University of Science and Technology of China, Suzhou, Jiangsu 215123, China. [3]CAS Key Laboratory of Low-Carbon Conversion Science and Engineering, Shanghai Advanced Research Institute, Chinese Academy of Sciences, Shanghai 201203, China. [4]School of Chemical Engineering, University of Chinese Academy of Sciences, Beijing 100049, China. [5]Key Laboratory of Functional Molecular Solids, Ministry of Education, Anhui Engineering Research Center of Carbon Neutrality, College of Chemistry and Materials Science, Anhui Normal University, Wuhu, Anhui 241000, China. [6]These authors contributed equally: Guangyu Liu, Yuan Zhong, Zehua Liu. ✉e-mail: gaoc@ustc.edu.cn; yjxiong@ustc.edu.cn

renewable electricity or solar light, are two widely explored routes to solar-driven chemical transformations[4–7]. Although significant progress has been made on chemical conversion of $CO_2$ to $C_1$ and $C_2$ products (e.g., CO, $CH_4$, HCOOH, $C_2H_4$, $CH_3CH_2OH$) through electrocatalysis or photocatalysis, it remains challenging to achieve sustainable synthesis of value-added food (e.g., long-chain sugars) by directly recycling $CO_2$. In comparison with electrocatalysis and photocatalysis, biocatalytic organisms offer the capability of producing food but require carbon oxygenates as feedstocks[8]. To this end, a feasible strategy for achieving artificial food synthesis is the integration of electrocatalytic $CO_2$ conversion modules, considering the substantially higher efficiency than photocatalysis, with biologically active components to combine their complementary advantages. Recently, breakthroughs have been made in developing chemical–biological hybrid systems to achieve food production, by coupling two-step $CO_2$ electrolysis with microorganism fermentation (glucose product)[9] or cultivation (yeast and mushroom-producing fungus)[10]. However, even by genetic engineering, the complex metabolic networks existing inside microbial cells pose a grand challenge to the customization of carbonaceous products. Moreover, the commonly used model heterotrophic microorganisms (e.g., *Escherichia coli*, *Saccharomyces*

*cerevisiae*) require sugars as necessary nutrients, which cannot fully achieve carbon neutrality[8,11]. Alternatively, according to diverse upstream products from chemical $CO_2$ conversion, the biological components in the hybrid system can be completely constructed in vitro by freely integrating different enzymatic reactions, which will be more favorable for customizable food production in contrast to in vivo metabolic pathway and have no consumption of carbonaceous nutrients[12,13].

Here, we report a hybrid electrocatalytic–biocatalytic flow system for achieving artificial synthesis of sugar directly from $CO_2$, by coupling photovoltaics-powered electrocatalysis with spatially separate enzyme cascade platform (Fig. 1a). Such a flow system in principle can be customized achieving efficient and renewable food production. Driven by a photovoltaic cell under simulated sunlight, a one-step electrocatalytic process efficiently converts $CO_2$ to formate over a bismuth-nanowire catalyst in a flow reactor. The produced formate solution can be directly fed into our designed tandem bioreactors, which comprise a five-enzyme cascade platform to achieve the efficient and selective conversion of formate into $C_6$ sugar—L-sorbose—a famous food additive as well as an important intermediate for industrial production of L-ascorbic acid. Such a hybrid electrocatalytic

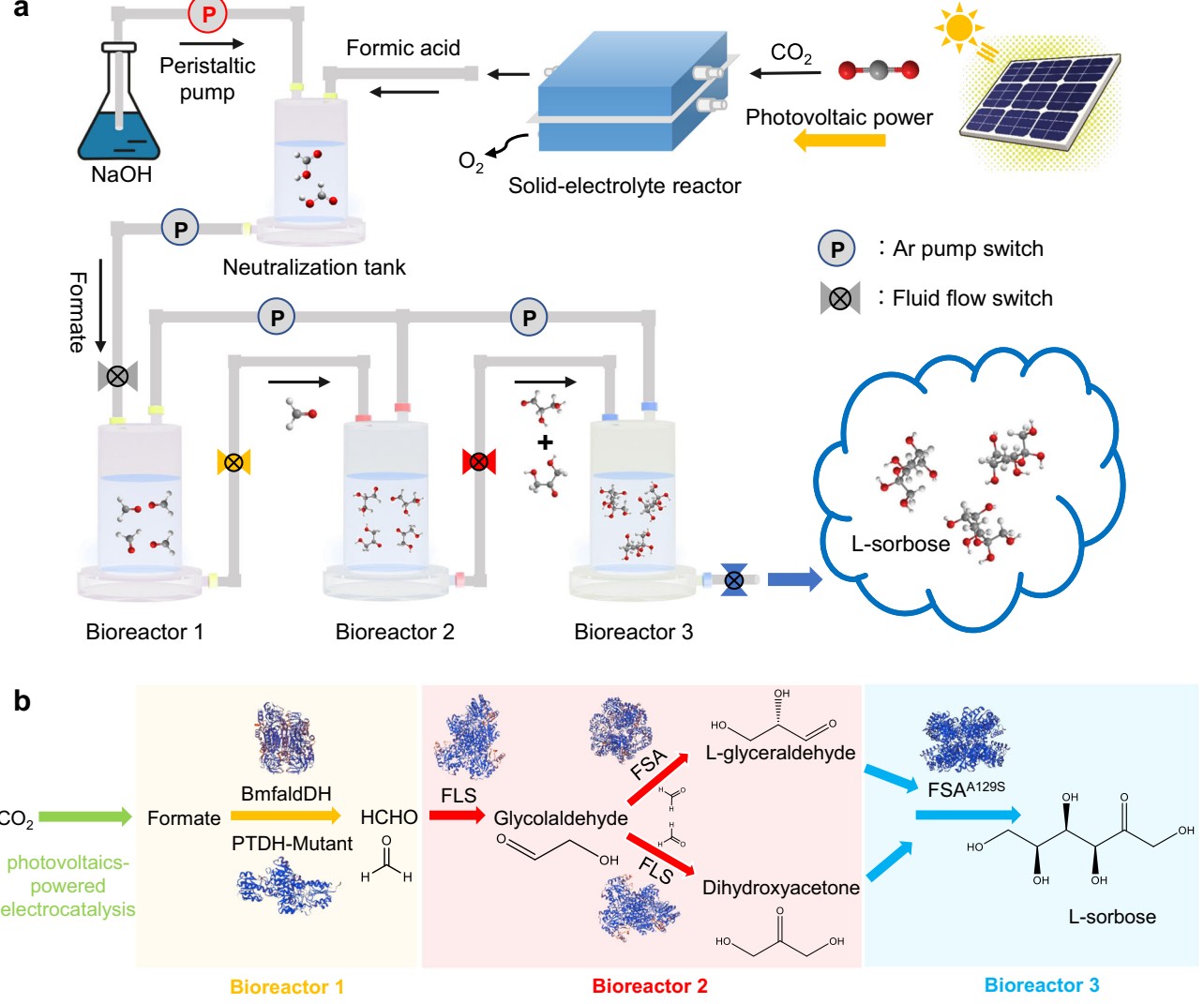

**Fig. 1 | Schematic illustration of the customizable electrocatalytic–biocatalytic flow system for solar-driven food production directly from $CO_2$.**
**a** Electrocatalytic–biocatalytic flow system, in which $CO_2$ was first converted into formate by photovoltaics-powered electrocatalysis, followed by direct injection of produced formate into designed tandem bioreactors for food production.
**b** Synthetic pathway for converting $CO_2$ to $C_6$ sugar—L-sorbose as a proof of concept.

−biocatalytic flow system for artificial food synthesis offers an excellent stability even after long-term continuous operation. Through screening out highly efficient enzymes via genetic mutation and bioinformatics, our designed system enables the conversion of $CO_2$ to sugar with a solar-to-food energy conversion efficiency of 3.5%, outperforming natural photosynthesis by over three-fold. This efficiency can be further improved by employing a matched photovoltaic cell with higher efficiency. With this hybrid solar-driven artificial synthesis system, it is anticipated that more enzymatic reactions can be integrated into the cascade enzyme platform in the future, achieving production of diverse foods directly from $CO_2$. This work provides a paradigm shift in food production, overcoming the limitations of traditional cultivation, which will particularly be available for applications in confined environments such as space station.

## Pathway design for $C_6$ sugar production from $CO_2$

As $CO_2$ is a stable linear molecule with a strong bond energy of 750 kJ/mol, it remains a grand challenge to directly activate C=O bonds by enzyme or microorganism with high efficiency[14]. Moreover, the overall efficiency of $CO_2$ conversion via biological approaches is restricted by mass transfer at the gas–liquid interface due to the low solubility of $CO_2$ in water[15]. Therefore, the approach that firstly achieves highly efficient conversion of $CO_2$ into a soluble liquid $C_1$ product (e.g., formic acid) by chemical electrocatalysis using gas diffusion electrode to overcome the mass transfer limitation, followed by the chemical transformations via biological enzyme cascade catalysis, is more feasible to achieve the food production. In principle, formaldehyde is a feasible feedstock for enzyme cascade reactions. Direct conversion of $CO_2$ to formaldehyde under mild reaction conditions can be realized through the photochemical, electrochemical and enzymatic approaches[16]. To minimize overall energy consumption, the conversion of $CO_2$ to formaldehyde powered directly by renewable light or electricity would be preferable. However, the state-of-the-art photocatalysts and electrocatalysts for $CO_2$ conversion encounter the limitations with both low yield and selectivity for producing formaldehyde, which is prone to be reduced or oxidized. In contrast, enzymatic reduction of $CO_2$ displays high selectivity toward formaldehyde production. Nevertheless, the enzymatic reduction of $CO_2$ is normally a multi-enzymatic and multistep reaction, in which the related enzymes have modest activity and require cofactors as electron donors. To combine the complementary advantages of various approaches, we chose electrocatalytic $CO_2$ conversion, considering the substantially higher efficiency, to produce formate as the starting feedstock, which was coupled with one-step enzymatic catalysis for formate-to-formaldehyde transformation to enable the following enzyme cascade reactions.

As a proof of concept, we designed an artificial enzymatic pathway for converting formate to $C_6$ sugar−L-sorbose (Fig. 1b). Specifically, the enzyme cascade process involves four steps: (i) The formate, generated by $CO_2$ electrocatalysis, is co-catalyzed by the basic local alignment search tool (BLAST)-screened formaldehyde dehydrogenase from *Burkholderia multivorans* (BmfaldDH) and genetically mutated phosphite dehydrogenase (PTDH) to obtain a high concentration of formaldehyde (HCHO); (ii) HCHO is transformed into $C_2$ product−glycolaldehyde, catalyzed by artificially designed aldolase (FLS). (iii) HCHO is further converted to $C_3$ products−dihydroxyacetone (DHA) and L-glyceraldehyde, which can be achieved via catalysis by FLS and D-fructose-6-phosphatase aldolase (FSA) from *Escherichia coli*, respectively[17–19]. (iv) DHA and L-glyceraldehyde are coupled to yield the final product−L-sorbose, catalyzed by A129S mutant of FSA ($FSA^{A129S}$)[20].

## Solar-driven electrocatalytic $CO_2$ conversion to formate

Under the synthetic roadmap, we first constructed a one-step electrocatalysis module for efficient conversion of $CO_2$ to formate. Bismuth (Bi)-based electrocatalysts are promising for such a conversion

due to their high selectivity (>90%), abundant reserves and environmental friendliness[21]. To this end, we prepared Bi nanowires (NWs) through a galvanic replacement method as electrocatalysts for formate production from $CO_2$ reduction[22]. Transmission electron microscopy (TEM) and scanning electron microscopy (SEM) images show that the as-prepared Bi NWs have uniform diameters of approximately 30−50 nm and lengths up to hundreds of nanometers, while X-ray diffraction (XRD) pattern and X-ray photoelectron spectroscopy (XPS) reveal that the Bi NWs possess a rhombohedral phase of metallic Bi (Supplementary Figs. 1–4). The as-prepared Bi NWs can directly serve as electrocatalysts for $CO_2$ conversion to formate.

To operate at conditions more relevant to an industrial electrolyzer and achieve optimal production, the electrocatalytic performance of Bi NWs for $CO_2$ reduction was evaluated using a three-electrode flow cell system with a gas diffusion electrode[23]. As shown in the schematic flow cell (Supplementary Fig. 5), Bi NWs catalyst were supported on a gas diffusion layer (GDL) to circumvent the mass transfer limitation due to the inherently low diffusion and solubility of $CO_2$ in aqueous solution. Electrocatalytic $CO_2$ reduction over Bi NWs was performed under different cathodic currents in 0.5 M $KHCO_3$ electrolyte, and the Faradaic efficiencies (FEs) and production rates of generated formate at different current densities are depicted in Fig. 2a. The Bi NWs electrocatalyst exhibit high FEs (> 92%) toward formate at a wide current window from 50 to 400 mA $cm^{-2}$, and achieve the optimal FE of 95.4% at a current density of 300 mA $cm^{-2}$. Such a catalytic performance was enhanced over commercial Bi nanopowders measured under identical conditions (Supplementary Figs. 6 and 7). To avoid the negative effect on downstream enzymatic catalysis and the anode degradation aroused by highly concentrated bicarbonate electrolyte, we further employed a porous solid-electrolyte reactor to directly produce pure HCOOH solution without containing bicarbonate (Fig. 2b, Supplementary Fig. 8), in which the $IrO_x$ supported on Ti foam was adopted as the anode. In such a system, the $HCOO^-$ generated from $CO_2$ reduction and $H^+$ from water oxidation were driven by electrical field cross the ion exchange membrane, and in turn, the generated HCOOH at the middle solid-electrolyte layer was then blown out by using humidified nitrogen gas. With the solid-electrolyte cell, the Bi NWs catalysts exhibited the FE > 80% for $CO_2$-to-HCOOH conversion under a current of 400 mA during 12 h of operation (Supplementary Fig. 9), offering the continuous and adequate supply of formic acid with a production rate of 6.40 mmol/h for the downstream enzyme catalysis. TEM, XRD and XPS characterizations of the catalyst after the electrocatalytic operation in solid-electrolyte reactor reveal no structural degradation of the Bi NWs catalysts (Supplementary Figs. 10–12), indicating the good stability of prepared catalysts.

To achieve light-driven $CO_2$ conversion, a photovoltaic cell with a certified solar-to-electricity efficiency of 15.5% (Supplementary Fig. 13, Supplementary Table 1) was implemented to harvest solar energy and power the electrocatalysis module. Illuminated with standard AM 1.5 G solar light intensity (100 mW $cm^{-2}$, 1 sun) at room temperature, the photovoltaic cell can generate an open-circuit voltage of 5.85 V and a short-circuit current of 157.3 mA (Fig. 2c). In addition, to achieve an efficient photovoltaic cell-powered electrocatalysis (PV-EC) system, the electrolyzer cell was operated at a voltage of 4.29 V with a current density of 3.57 mA $cm^{-2}$, which is close to the maximum power point of the solar cell. Through integrating the photovoltaic cell with electrocatalysis module, this solar-driven one-step $CO_2$ electrolysis offers a HCOOH production rate of 2.04 mmol $h^{-1}$ without external bias (Fig. 2d). With 120 h of continuous electrolysis, the PV-EC system shows a negligible decrease in the FE of HCOOH, operating current and cell voltage (Fig. 2e), indicating the excellent stability for continuous and efficient conversion of $CO_2$ to HCOOH by our constructed PV-EC module. The related products during the 120 h solar-driven electrocatalytic $CO_2$ conversion include HCOOH (average FE: 82.2%), $H_2$ (average FE: 12.8%) and CO (average FE: 2.7%) (Supplementary

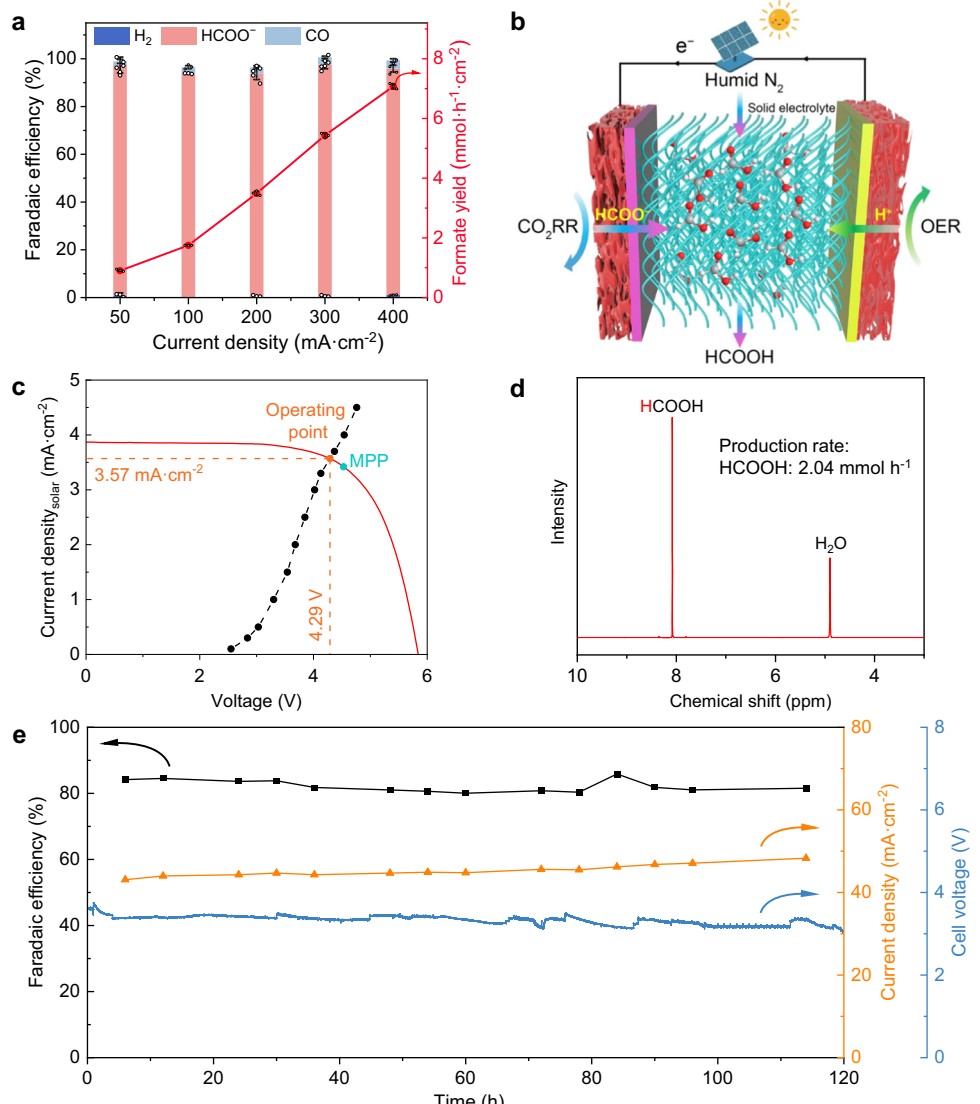

**Fig. 2 | Solar-driven one-step electrocatalytic CO₂ conversion to formic acid.**
**a** Faradaic efficiencies of products and the formate production rates over the prepared Bi NWs catalyst at different current densities in flow cell with a gas diffusion electrode. Data points are reported as mean ± standard deviation derived from 3 independent experiments ($n = 3$). **b** Schematic illustration of $CO_2$ reduction to HCOOH in a porous solid-electrolyte reactor, powered by a photovoltaic cell (with an illuminated surface area of 38.9 cm²). **c**, Photovoltaic and electrocatalytic J–V behaviors. The J–V curve of solar cell (red line) was measured under simulated solar irradiation. The measured operating current densities of the electrolyzer at different voltages have been marked by black dots. The cyan dot represents the maximum power point (MPP) of the solar cell, and the orange dot represents the operating point (intersection of the two curves) of solar-driven electrocatalytic $CO_2$ conversion, with the orange dashed lines showing the corresponding voltage and current density normalized by the illuminated surface area. All the potentials were measured without iR compensation. **d** ¹H NMR spectrum for produced HCOOH over Bi NWs electrocatalyst in the solid-electrolyte reactor powered by a photovoltaic cell. **e** Long-term operation test of $CO_2$ reduction to pure HCOOH solution by using our constructed porous solid-electrolyte reactor powered by a photovoltaic cell. Source data are provided as a Source Data file.

Figs. 14 and 15). The single-pass outlet formic acid concentration is 1.1 M, and can be tuned easily by the amount of water in the collecting bottle. Simultaneously, a 1:1 stoichiometric sodium hydroxide solution can be pumped by peristaltic pump into the obtained formic acid solution to obtain a 200 mM formate solution. Eventually, the obtained formate solution was directly fed into the downstream bioreactors for biocatalytic reactions. It is worth mentioning that formate is the only product in liquid phase so that the direct feeding of electrocatalytic $CO_2$ reduction product into enzyme cascade reactions becomes feasible.

## Enzyme engineering for optimizing formate-to-formaldehyde conversion

The efficient generation of formate provides a sustainable feed for downstream reactions. As formaldehyde is the fundamental building

blocks for sugar production in the enzyme cascade reactions, the conversion of formate to formaldehyde, as the bridge linking chemical and biological catalytic modules, holds the key to determining the overall efficiency of the entire system for sugar production. Formaldehyde dehydrogenase (FaldDH) can catalyze the conversion of formate to formaldehyde with reduced nicotinamide adenine dinucleotide (NADH) as cofactor to provide reducing power[24]. As this conversion is a reversible reaction[25], the sufficient supply of NADH is essential for the unidirectional catalysis toward formaldehyde production. In this regard, both the selection of FaldDH and regeneration of NADH were optimized to achieve the maximum formate-to-formaldehyde conversion. To optimize the selection of FaldDH, we screened 29 FaldDHs and evaluated their formate reduction ability by using bioinformatics—the BLAST method. Based on their performance

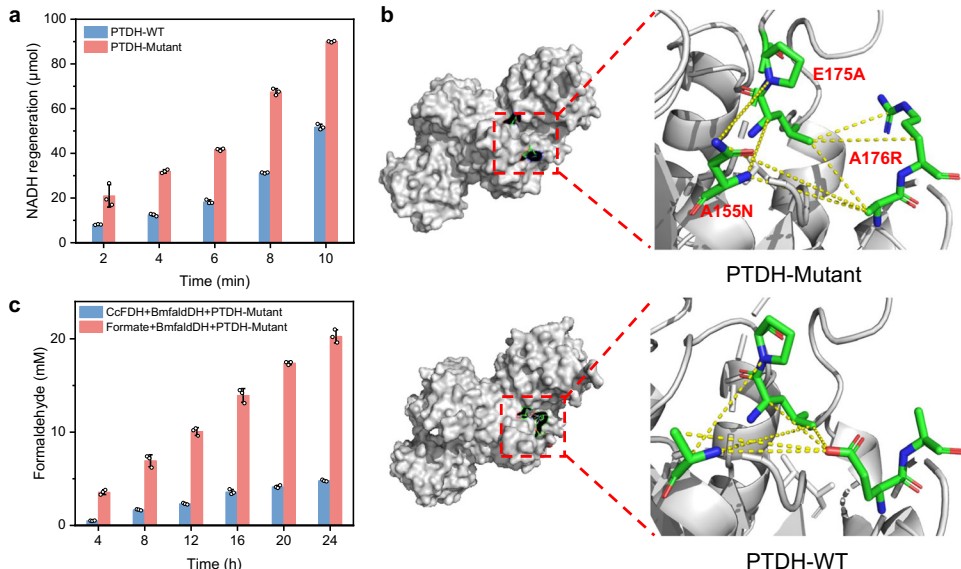

**Fig. 3 | Enzyme engineering for optimizing formate-to-formaldehyde conversion. a** The amount of NADH regenerated over wild-type PTDH (PTDH-WT) and PTDH after targeted mutation (PTDH-Mutant) with a concentration of 2 mg/mL. **b** Homology modeling of PTDH-WT and PTDH-Mutant with amplification of amino acids at A155N, E175A and A176R. Data points are reported as mean ± standard deviation derived from 3 independent experiments (*n* = 3). **c** The yield of

formaldehyde by coupling enzyme-catalyzed formate-to-formaldehyde conversion module with light-driven electrocatalytic $CO_2$-to-formate conversion module or enzymatic CcFDH-catalyzed $CO_2$-to-formate conversion module. Data points are reported as mean ± standard deviation derived from 3 independent experiments (*n* = 3). Source data are provided as a Source Data file.

in crude cell lysates, we used the phenazine methanesulfonate (PMS)−nitroblue-tetrazolium (NBT) system[24] to construct a library of 29 FaldDHs. Through analyzing the formate-reducing ability of these FaldDHs (Supplementary Table 2), the results show that formaldehyde dehydrogenase from *Burkholderia multivorans* (BmfaldDH, Supplementary Fig. 16) is the most active enzyme among them for formate-to-formaldehyde conversion. Additionally, we used isothermal titration calorimetry (ITC) to measure the thermodynamic data for evaluating the interaction of BmfaldDH with formate and NADH (Supplementary Fig. 17). The result indicates that BmfaldDH is thermodynamically capable of catalyzing the formate-to-formaldehyde conversion.

We further optimized the NADH regeneration by attempting electrocatalytic[26] and enzymatic methods. It turns out that the enzymatic method using phosphite dehydrogenase (PTDH) from *Pseudomonas sp. K*[27] offers higher efficiency for regeneration of NADH in this work (Fig. 3a, Supplementary Fig. 18), whereas the electrocatalytic method offers a promising way for NADH regeneration without the requirement of phosphite[28]. To further enhance the capacity of PTDH for NADH regeneration, we performed a targeted mutation of PTDH at A155N, E175A and A176R. The constructed mutant of PTDH (denoted as PTDH-Mutant) achieves a substantial enhancement of NADH regeneration, outperforming the wild-type PTDH by approximately two-fold (Fig. 3a, Supplementary Fig. 19). Even at a low concentration, this mutant of PTDH enables the rapid NADH regeneration (Supplementary Fig. 20), demonstrating its excellent capacity in supplying sufficient NADH for BmfaldDH toward efficient formate-to-formaldehyde conversion. Additionally, to enable a close connection between the electrochemical and biological catalytic modules, the experimental conditions such as pH, temperature, substrate concentration and the ratios of BmfaldDH and PTDH-Mutant were optimized by using response surface methodology (Supplementary Fig. 21). Indeed, with PTDH-Mutant for NADH regeneration, BmfaldDH demonstrates efficient formaldehyde production from formate, achieving a formaldehyde concentration of 6.26 mM after 3 h reaction (Supplementary Fig. 22). This enzymatic system also offers a significantly higher efficiency toward formaldehyde production in

various buffers in contrast to that by using wild-type PTDH (Supplementary Fig. 23), demonstrating the versatility in integrating various up- and downstream catalytic modules.

To further understand the enhanced catalytic capacity of PTDH mutant for NADH regeneration, the roles of the three mutation sites were illustrated by designing the three PTDH with single site mutations (A155N, E175A and A176R, Supplementary Fig. 24), followed by the determination of their kinetic parameters (Supplementary Table 3) and the homology modeling analysis of PTDH based on protein sequence with targeted mutations (Fig. 3b, Supplementary Figs. 25−27). The kinetic parameters show that the three single-site mutants (denoted as PTDH-A155N, PTDH-E175A and PTDH-A176R) can enhance the capacity of PTDH for NADH regeneration to varying degrees compared to the wild-type PTDH (PTDH-WT). The homology modeling analysis reveals that, compared to wild-type PTDH, the A155N mutation in the PTDH mutant promotes the binding of $NAD^+$ by strengthening the salt bridge with phosphate groups (Supplementary Fig. 27)[29,30], which is verified by the reduced $K_m$ value of PTDH-A155N in contrast to PTDH-WT. The E175A mutation can replace electronegative Glu175 residues (alanine, glycine, and valine) with uncharged amino acid residues to reduce the electrostatic repulsion to phosphate groups[30], while the A176R mutation replaces electronegative Ala176 with an electropositive amino acid that can interact with the phosphate of $NAD^+$ statically and simultaneously bind to the E175A mutant base by hydrogen bonding[31,32], thus further enhancing the capacity of PTDH to bind NADH. These impacts of the two mutations are supported by the kinetic parameters of PTDH-E175A and PTDH-A176R (Supplementary Table 3). The $K_m$ values of E175A and A176R mutants are both smaller than that of PTDH-WT, while the $k_{cat}$ values of the E175A and A176R mutants increase significantly in contrast to PTDH-WT. Taken together, these improvements by target mutations contribute to a significant enhancement in NADH regeneration over PTDH mutant.

Upon achieving the optimal formate-to-formaldehyde conversion, we then combined the light-driven electrocatalytic $CO_2$-to-formate conversion module with the enzyme-catalyzed formate-to-formaldehyde conversion module to verify the feasibility of the

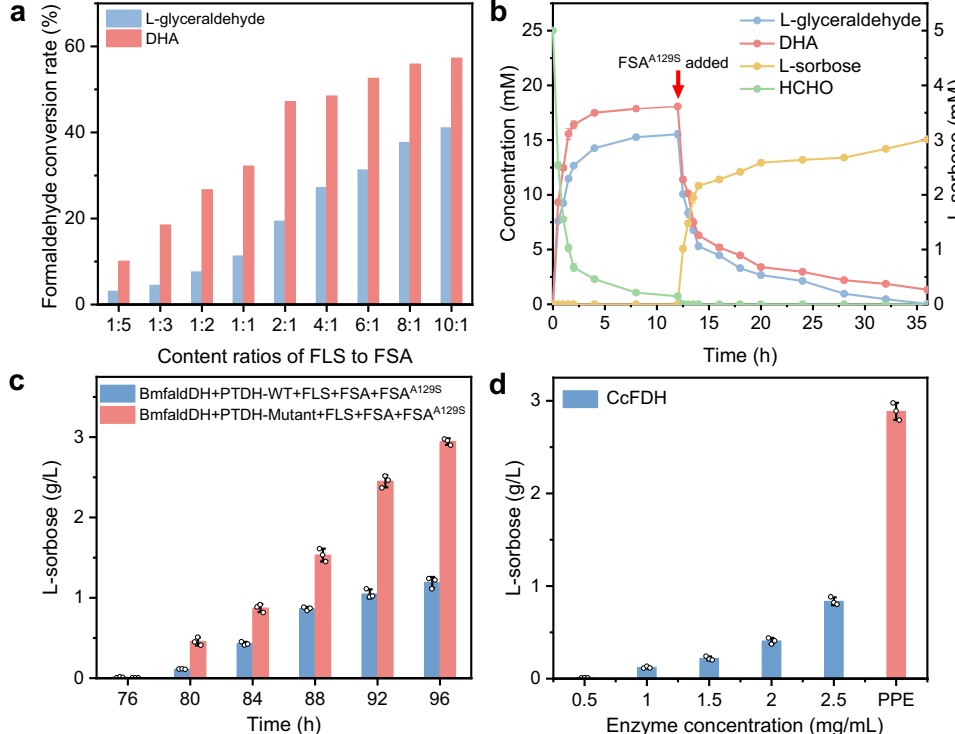

**Fig. 4 | Enzyme cascade for L-sorbose production. a** The conversion rates of formaldehyde toward production of DHA and L-glyceraldehyde with various content ratios of FLS to FSA. **b** The time-dependent concentrations for formaldehyde, DHA, L-glyceraldehyde, FSA^A129S and L-sorbose during the enzyme cascade for L-sorbose production from formaldehyde. **c** The yields of L-sorbose in the one-pot enzyme cascade for formate-to-L-sorbose conversion with wild-type PTDH (PTDH-WT) and mutant of PTDH (PTDH-Mutant). Data points are reported as mean ± standard deviation derived from 3 independent experiments (n = 3). **d** The yield of L-sorbose by coupling enzyme-catalyzed formate-to-L-sorbose conversion module with light-driven electrocatalytic $CO_2$-to-formate module or enzymatic CcFDH-catalyzed $CO_2$-to-formate module after 24 h reaction. PPE photovoltaics-powered electrocatalysis. Data points are reported as mean ± standard deviation derived from 3 independent experiments (n = 3). Source data are provided as a Source Data file.

coupled biotic−abiotic hybrid system for light-driven formaldehyde production directly from $CO_2$. The formate generated by electrocatalysis was injected into the BmfaldDH−PTDH-Mutant enzymatic system using a peristaltic pump. Driven by standard 1 sun solar light, this coupled system offers a concentration of 21 mM for produced formaldehyde after 24 h of reaction (Fig. 3c, Supplementary Fig. 28), providing a sufficient and sustainable feed for downstream enzyme cascade module toward sugar production. This hybrid electrocatalytic−biocatalytic system even offers enhanced $CO_2$-to-formaldehyde conversion, in contrast to the complete enzymatic system that converts $CO_2$ to formate using formic acid dehydrogenase from *Clostridium carboxidivorans P7T* (CcFDH) with NADH as cofactor[33] (Fig. 3c, Supplementary Figs. 16 and 29). This further demonstrates the superiority of our design integrating electrocatalytic $CO_2$ conversion module with biologically active components. Such an enzyme engineering that can rationally optimize formate-to-formaldehyde conversion paves the way to the following enzyme cascade system toward $C_6$-sugar production.

**Enzyme cascade for L-sorbose production**

Given that efficient $CO_2$-to-formaldehyde conversion has been achieved, we then sought to construct the enzyme cascade module for converting formaldehyde (HCHO) to C6 sugar−L-sorbose. Following our synthetic roadmap (Fig. 1b), FLS and FSA were constructed and purified to perform the HCHO-to-$C_3$ product conversion (Supplementary Fig. 30). The results show that FLS can catalyze the condensation of HCHO to sequentially produce glycolaldehyde ($C_2$) and DHA ($C_3$), achieving a separation yield of 90% toward DHA after 12 h reaction (Supplementary Figs. 31−33). Interestingly, when FLS collaboratively works with FSA, HCHO can be sequentially converted into glycolaldehyde ($C_2$) and then into L-glyceraldehyde ($C_3$). However, with the 1:1 content ratio of FLS to FSA, the conversion rate from HCHO to DHA is significantly higher than that to L-glyceraldehyde, leading to an unbalanced proportion for following production of $C_6$ sugar. Essentially, the glycolaldehyde intermediate adsorbed on FLS is more readily further converted to DHA, rather than is desorbed from FLS and captured by FSA to produce L-glyceraldehyde[34]. In this regard, the content ratios of FLS to FSA were optimized to achieve a matched output for DHA and L-glyceraldehyde. When the ratio of FLS to FSA is 10:1, the conversion rates toward DHA and L-glyceraldehyde are 53.3% and 41.2%, (Fig. 4a, Supplementary Fig. 34), respectively, providing a suitable feed ratio for the subsequent L-sorbose production.

To synthesize the final $C_6$ sugar, A129S mutant of FSA (FSA^A129S) was constructed to catalyze the aldol addition reaction of DHA and L-glyceraldehyde (Supplementary Figs. 35−37). As revealed by kinetic parameters, FSA^A129S has an obviously lower $K_m$ (Michaelis constant) and a higher $k_{cat}$ (turnover number) value toward DHA (Supplementary Table 3). This indicates that the A129S mutation on FSA can enhance the affinity and catalytic efficiency toward DHA, thus allowing the efficient aldol addition[35]. As a result, the mutant FSA^A129S can catalyze the addition reaction of DHA ($C_3$) and L-glyceraldehyde ($C_3$) to produce L-sorbose ($C_6$), achieving a separation yield of 93% after 12 h reaction (Supplementary Fig. 38). After optimizing the content ratio of FLS to FSA and verifying the individual catalytic capacity of FLS, FSA and FSA^A129S, we then performed the enzyme cascade for L-sorbose production from HCHO by the combination of FLS, FSA and FSA^A129S (Fig. 4b, Supplementary Fig. 39). As shown in Fig. 4b, the concentration of HCHO decreases dramatically within the initial 12 h reaction along with the gradual increase in the concentration of DHA and L-glyceraldehyde, indicating the efficient conversion of HCHO to DHA and

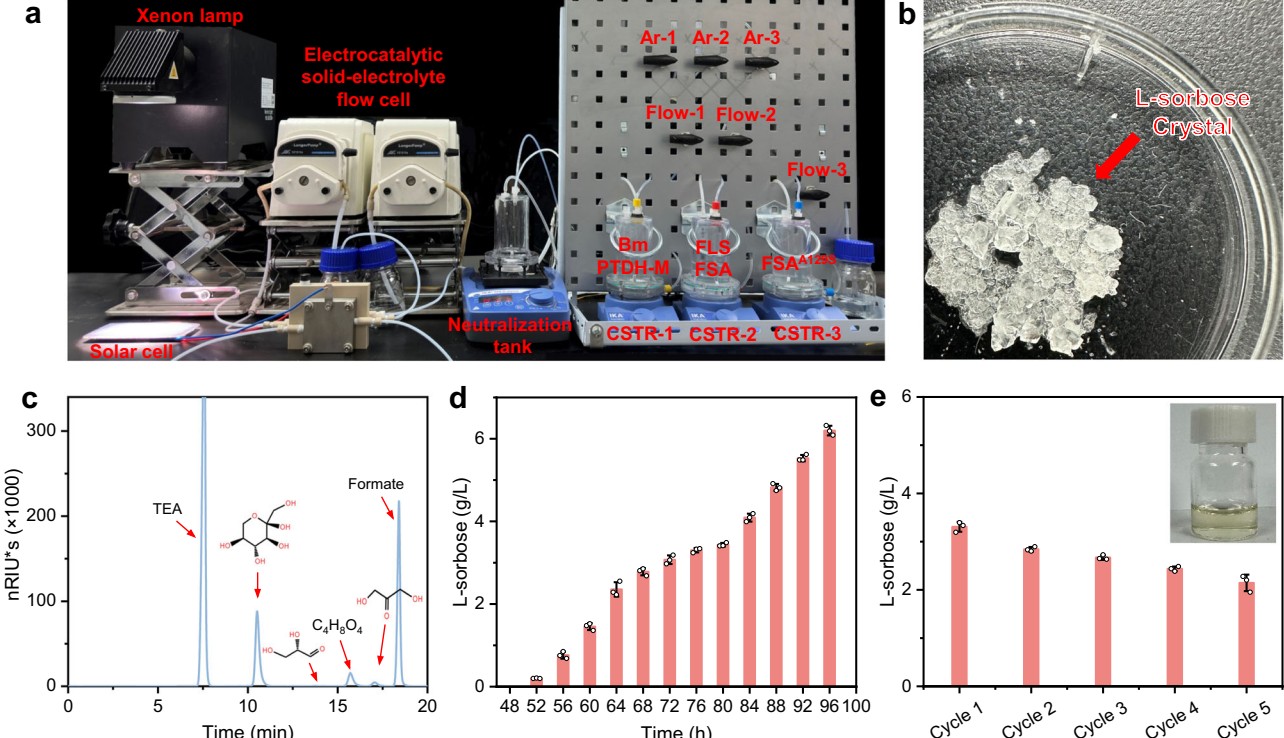

**Fig. 5 | Solar-driven sugar production from CO₂ in electrocatalytic–biocatalytic flow system. a** Photograph of the designed electrocatalytic–biocatalytic flow system powered by a photovoltaic cell. **b** Photograph of the obtained high-purity L-sorbose crystals. **c**, High-performance liquid chromatography analysis of final reaction solution in the solar-driven electrocatalytic–biocatalytic flow system. **d** The time-dependent yield for L-sorbose in the solar-driven electrocatalytic –biocatalytic flow system. Data points are reported as mean ± standard deviation derived from 3 independent experiments (*n* = 3). **e** Cycling tests for the solar-driven electrocatalytic–biocatalytic flow system. Each cycle takes 24 h. Inset shows the final condensed reaction solution containing L-sorbose after reduced pressure distillation. Data points are reported as mean ± standard deviation derived from 3 independent experiments (*n* = 3). Source data are provided as a Source Data file.

L-glyceraldehyde. After 12 h of the reaction, the addition of FSA^A129S arouses the sharp decrease in the concentration of DHA and L-glyceraldehyde, while the concentration of L-sorbose increases accordingly, achieving a high separation yield of 72%. This demonstrates the validity of the enzyme cascade module for L-sorbose production.

Upon verifying the validity of separated enzymatic formate-to-HCHO and HCHO-to-L-sorbose conversion, we then sought to connect the two separated enzymatic modules to verify the whole enzyme cascade for formate-to-L-sorbose conversion. The combination of BmfaldDH, PTDH-Mutant, FLS, FSA and FSA^A129S enables the efficient production of L-sorbose from formate (Fig. 4c, Supplementary Fig. 40). Moreover, this enzymatic system offers a significantly higher efficiency toward L-sorbose production in contrast to that by using wild-type PTDH, further confirming the essential role of NADH regeneration in determining the efficiency of entire enzymatic system. Through the further coupling with electrocatalytic CO₂-to-formate conversion module, the constructed hybrid electrocatalytic–biocatalytic system endows a prominently enhanced L-sorbose yield in contrast to the complete enzymatic system by using CcFDH even at a high concentration of 2.5 mg/mL (Fig. 4d), illustrating the great advantages in integrating electrocatalytic CO₂ conversion module with biocatalytic modules.

**Solar-driven L-sorbose production from CO₂ in flow cascade reactors**

To further validate the possibility of scaling up our constructed system toward practical application, we then designed flow cascade bioreactors to implement the constructed electrocatalytic–biocatalytic system toward continuous solar-driven sugar production directly from CO₂ (Fig. 5a, Supplementary Fig. 41). In each individual continuous

stirred tank reactor (CSTR), the enzymes are separated from the products using ultrafiltration membranes with suitable molecular filtration capacity, which not only avoids the interference on downstream reactions but also preserves the enzymes in individual CSTR for continuous reaction. The liquid flow between different CSTRs was controlled by Ar gas pump using adjustable valves. The biocatalytic flow cascade system is composed of three CSTRs: the first CSTR contains BmfaldDH, PTDH-Mutant and NAD⁺ to implement formate-to-HCHO conversion; the second CSTR contains FLS and FSA with pre-adjusted content ratio to implement HCHO-to-C₃ (DHA and L-glyceraldehyde) conversion; the third CSTR contains FSA^A129S to implement addition reaction of DHA and L-glyceraldehyde toward L-sorbose production. Note that to avoid the negative effect on downstream enzymatic catalysis by highly concentrated bicarbonate electrolyte (Supplementary Fig. 42), a porous solid-electrolyte reactor has been employed to directly produce pure HCOOH solution, followed by adjusting the pH with cost-effective NaOH.

Upon light irradiation, the high concentration of pH-adjusted formate solution generated by the light-driven electrocatalytic CO₂ conversion module was directly fed into CSTR-1 for biocatalytic formate-to-HCHO conversion using a peristaltic pump. Benefited from the high efficiency of PTDH-Mutant for NADH regeneration, the concentration of NADH remains a high level in CSTR-1 during the reaction (Supplementary Fig. 43), suggesting that BmfaldDH can convert formate to formaldehyde constantly with sufficient phosphite. Then, the reaction medium in CSTR-1 containing the generated high-concentration HCHO with 24 h reaction was pumped into CSTR-2 for production of DHA and L-glyceraldehyde with an approximately equivalent yield. Simultaneously, the reaction medium in CSTR-2 containing the generated high-concentration DHA and

L-glyceraldehyde with 24 h reaction was pumped into CSTR-3 for L-sorbose production. The final reaction medium in CSTR-3 with 24 h reaction was pumped into a conical flask and collected for concentration and crystallization (Supplementary Fig. 44). After reduced pressure distillation and vacuum drying, high-purity L-sorbose crystals were obtained (Fig. 5b, Supplementary Fig. 45), demonstrating the excellent efficiency for sugar production by our electrocatalytic−biocatalytic flow system. Note that the second run only needs 24 h reaction to obtain the same L-sorbose production as CSTR-1 and CSTR-3 can operate simultaneously with CSTR-3. The results of full carbon and metabolite analysis in each CSTR are summarized in Supplementary Table 4.

The residual formate solution has no significant influence on the downstream enzymatic reactions in CSTR-2 and CSTR-3 (Supplementary Fig. 46). More importantly, the residual formate in CSTR-3 can be converted to L-sorbose with a similar efficiency after being collected and fed into CSTR-1 again (Supplementary Fig. 47), indicating that our flow system can fully convert formate to L-sorbose and thus avoid the carbon loss to a great extent. In addition, no significant amount of methanol is derived from the formaldehyde Cannizzaro disproportionation reaction in CSTR-1 after 24 h of reaction, and thus the product yields in each CSTR are not significantly influenced (Supplementary Figs. 48 and 49). Under 1 sun light intensity, a high yield of 105.0 mg/L/h of L-sorbose was obtained from the second cycle (Fig. 5c and d), achieving an exciting solar-to-food energy conversion efficiency of 3.5% that outperforms natural photosynthesis by over three-fold (see Supplementary Information for detailed calculation). Furthermore, this hybrid flow system offers a relatively high stability in the durability test, maintaining 79.5% of the initial yield for L-sorbose even after 5 consecutive cycles with a total reaction time of 120 h for each CSTR (Fig. 5e), demonstrating the sustainability of the system. It is worth noting that although additional $Na_2HPO_3$ was required, our photovoltaics-powered electrocatalytic−biocatalytic flow system is economically feasible from the perspective of product value. The production of 1 Kg L-sorbose requires 6.1 Kg $Na_2HPO_3$, which is converted to 7.9 Kg $Na_3PO_4$ after providing electrons. Note that the price value of $Na_2HPO_3$, $Na_3PO_4$ and L-sorbose are about 1.8, 1.2 and 27.9 \$/Kg, respectively. Moreover, NADH can also be regenerated by electrochemical method, which offers the opportunity of avoiding using phosphite in the future. The above results demonstrate that our electrocatalytic−biocatalytic flow system has great potential for practical application.

To summarize, we have demonstrated an intriguing hybrid electrocatalytic−biocatalytic flow system, coupling photovoltaics-powered electrocatalytic module with biocatalytic enzyme cascade module engineered via genetic mutation and bioinformatic analysis, which offers sustainable, customizable, scalable and carbon-neutral access to light-driven sugar production directly from $CO_2$. Under standard solar light irradiation with $CO_2$ as the only carbon source, this flow system achieves conversion of $CO_2$ to high-purity edible L-sorbose with the yield of 105.0 mg/L/h and solar-to-sugar energy conversion efficiency of 3.5%, outperforming natural photosynthesis by over three-fold. This flow system can in principle be programmed by integrating diverse enzymes toward production of multifarious food (e.g., sugars, nutriments) from $CO_2$. For instances, if FSA is removed from the enzyme cascade system, $FSA^{A129S}$ can catalyze DHA to obtain the rare and expensive $C_4$ sugar−L-erythrulose[36]. Furthermore, as an important intermediate in traditional "two-step fermentation" process for industrial production of L-ascorbic acid[37], the produced L-sorbose from $CO_2$ allows the direct production of L-ascorbic acid directly from $CO_2$. Our approach to food production will particularly be available for applications in confined environments and physical space, such as space station or the region with atrocious circumstance or finite agricultural lands on earth. In addition, beyond being driven by light,

our flow system could also be powered by stored electricity or driven by other renewable electricity such as wind turbines, thus providing great potential for widespread and large-scale practical application. This work provides great opportunities for revolutionizing traditional farming and building a sustainable carbon-neutral food manufacturing industry.

## Methods

### Screening of formate-reducing FaldDHs
To screen FaldDHs of formate assimilated microorganisms and choose the optimal FaldDHs, a BLAST-based sequence comparison and systematic sequence screening approach was performed to evaluate the potential of formate-reducing homologs using the PpFaldDH sequence as the driver sequence. Sequence analysis led to the selection of 70 FaldDHs for further screening. A phenazine methosulfate (PMS) −nitroblue tetrazolium (NBT) system was then applied to detect the reduction activity of crude cell lysates of bacterial cells containing FaldDHs. Each 250 μL contained 5 μL of crude cell lysate, 235 μL of buffer solution, and 10 μL of reagent solution (300 μM NADH, 300 μM HCOONa, 300 μM NBT, and 30 μM PMS in the diluted form). The reaction was started by adding the reagent solution to the reaction mix containing a specific FaldDH and buffer solution, and the absorbance was then monitored at OD580 for up to 30 min in a 96-well plate. NBT, in the presence of PMS, reacted with the residual NADH to produce a blue-purple formazan. Based on the PMS−NBT screening results, a library of 29 FaldDHs was constructed. Finally, the library of 29 FaldDHs was analyzed for formate reduction or FaldDH assay.

### Cloning, expression and purification of CcFDH, BmfaldDH, PTDH-WT, PTDH-A155N, PTDH-E175A, PTDH-A176R, PTDH-Mutant, FLS, FSA and FSA^{A129S}
The CcFDH (UniProt E2IQB0), BmfaldDH (UniProt J4QK49), and PTDH-WT (UniProt F2YGD2) genes were cloned into pET28(a) using NdeI and XhoI, or BamHI and XhoI, or BamHI and NotI, respectively. The FLS and FSA (UniProt P78055) gene was cloned into pET21(a) using NdeI and HindIII, or NdeI and HindIII, respectively. DNA sequencing confirmed the cloned CcFDH, BmfaldDH, PTDH-WT, FLS and FSA genes free from point mutations (General bio, China, Supplementary Table 5). To obtain PTDH-A155N, PTDH-E175A, PTDH-A176R, PTDH-Mutant and $FSA^{A129S}$ gene, site-directed mutagenesis of the resulting plasmid using the PCR-based QuikChange Method (Agilent Technologies) was used to introduce the A155N, E175A, and A176R or A129S substitutions into the PTDH coding sequence and FSA coding sequence, respectively[20,31]. DNA sequencing confirmed the cloned PTDH-A155N, PTDH-E175A, PTDH-A176R, PTDH-Mutant and $FSA^{A129S}$ genes to be right point mutations (General bio, China). Based on the codon preference of Escherichia Coli BL21(DE3), the target gene was codon optimized to facilitate the expression of the exogenous gene. The recombinant plasmids were transformed into competent Escherichia coli BL21(DE3) cells (Supplementary Table 6 and 7). Expression and purification of these genes were performed as previously described[33]. The recombinant CcFDH and BmfaldDH were expressed using 0.1 mM isopropyl-β-D-thiogalactopyranoside (IPTG) at 16 °C, while the recombinant PTDH, PTDH-A155N, PTDH-E175A, PTDH-A176R, PTDH-Mutant, FLS, FSA and $FSA^{A129S}$ were all expressed using 0.5 mM IPTG at 16 °C. To obtain the corresponding enzymes, bound Ni-NTA resins were washed with 10 mL of wash buffer (50 mM Tris, pH 7.4, 500 mM NaCl, 50 mM imidazole). The collected enzymes were then subjected to ultrafiltration, dialysis and desalination. The purified proteins were concentrated and stored with buffer (50 mM Tris-HCl pH 7.4, 150 mM NaCl). The purity of the enzymes was determined by sodium dodecyl sulfate-polyacrylamide gel electrophoresis (SDS-PAGE, Supplementary Figs. 16, 24 and 30) on 12% gels and visualized by staining with Coomassie Blue R-250 (Bio-Shop, Burlington, Canada, Supplementary Table 8).

## Electrochemical measurements

The electrochemical measurements were controlled by an electrochemical workstation (CHI 660e) equipped with a current amplifier (CHI 680c) or a CS310M electrochemical workstation (Wuhan CorrTest Instrument Co. Ltd.). For the flow-cell test, 6.25 mg of the Bi NWs was added into a mixture of isopropanol (970 μL) and Nafion ionomer solution (5%, Sigma–Aldrich) (30 μL), and dispersed by sonication to form a homogeneous ink. The catalyst ink was then dropped onto a gas diffusion layer (YLS 30T, Fuel Cell Store) as the cathode electrode (1 × 1 cm$^2$) with a mass loading of ~0.5 mg cm$^{-2}$. The high-purity $CO_2$ (Linde, 99.999%) gas was fed to the cathode with a constant flow rate (50 sccm) monitored by a mass flow controller (D08-1F, Sevenstar). A saturated Ag/AgCl electrode and a piece of Ni foam were adopted as the reference electrode and counter electrode, respectively. A piece of Nafion 117 membrane (Fuel Cell Store) was sandwiched between the anode and cathode. The electrolyte (0.5 M $KHCO_3$) was circulated in the anolyte and catholyte chambers at a flow rate of 1 mL/min during $CO_2$ electrolysis.

The membrane electrode assembly (MEA) electrolyzer with solid-state electrolyte (AmberChrom 1 × 8 chloride form) was also exploited to produce high concentrations of formic acid. In this case, a piece of GDL (2 × 2 cm$^2$) loading Bi NWs (0.5 mg cm$^{-2}$) or $IrO_x$ supported on Ti foam (1 mg cm$^{-2}$) was employed as the cathode and anode, respectively. The thickness of the interlayered solid-electrolyte is 2.0 mm. The thicknesses of the electrode gasketing along with Ti foam and carbon paper are 1.0 mm (with a Ti foam thicknesses of 1.0 mm) and 0.30 mm (with a carbon paper thicknesses of 0.23 mm), respectively. The cathode side was supplied with humidified $CO_2$ gas (20 mL min$^{-1}$), and $H_2SO_4$ (0.5 M) aqueous solution was circulated around the anode side. The internal resistance of the solid-electrolyte reactor is 2.91 Ω, which was measured via potentiostatic electrochemical impedance spectroscopy at frequencies ranging from 0.1 Hz to 1 MHz. To wash out possible trace impurities, the porous solid-electrolyte reactor was first stabilized for 60 min before liquid product collection. The generated HCOO$^-$ in the cathode and H$^+$ in the anode entered the porous solid-state electrolyte through an anion-exchange membrane (Sustainion X37-50 Grade 60) and a cation exchange membrane (Nafion 117), respectively. The HCOOH within the solid-state electrolyte layer was blown out using humidified nitrogen gas. All the potentials were measured without iR compensation.

The cathodic evolved gaseous products were quantified by gas chromatography. CO was analyzed by gas chromatograph (GC, 7890A, He carrier, Agilent) equipped with a flame ionization detector. $H_2$ was detected by gas chromatograph (GC, 7890B, Ar carrier, Agilent) equipped with a thermal conductivity detector. The cathodic liquid products were analyzed by $^1H$ nuclear magnetic resonance (NMR) spectroscopy (Bruker AVANCE AVIII 400). The Faradaic efficiency (*FE*) of $CO_2$ electrolysis products was calculated using the following equation:

$$FE(\%) = \frac{Q}{Q_{total}} \times 100\% = \frac{n_e \times n \times F}{Q_{total}} \times 100\%$$

where $Q$ and $Q_{total}$ represent the charges transferred into the corresponding product and the total charge passed through the cathode during electrolysis, respectively; $F$ represents the Faraday constant (96485 C/mol); $n$ is the mole amount of the corresponding product; and $n_e$ is the number of electrons transferred.

## Light-driven production of L-sorbose from $CO_2$ in flow cascade reactors

The light-driven electrocatalysis from $CO_2$ to formic acid was performed with a 300 W Xenon lamp (Perfect light, Beijing, China) coupled with an AM 1.5G filter illuminating on a commercial solar panel, which is connected by a cable to the electrocatalytic module. Subsequently, the formic acid and NaOH solution were pumped into a neutralization tank to form a formate solution. As our PV-EC system can generate 1.8 mL formic acid solution (1.1 M) per hour, the NaOH solution (340 mM) was pumped into the neutralization tank at a rate of 6 mL/h. Simultaneously, to provide a similar buffer system for downstream enzyme reactions, TEA buffer (0.5 M, pH 7.2) was added into the neutralized tank at a rate of 2 mL/h. Then, the produced formate solution was pumped by peristaltic pump into CSTR-1, which contained 150 mM sodium phosphite, 20 mM NAD$^+$, BmfaldDH (50 U) and PTDH-Mutant (10 U) in 100 mM TEA buffer with a pH of 7.2 under 27 °C and 120 rpm. The reaction medium in CSTR-1 with 24 h reaction was pumped by Ar into CSTR-2, which contained FLS (≥20 U) and FSA (≥ 50 U) with pre-adjusted content ratio, 1 mM $MgSO_4$ and 0.1 mM TPP in TEA buffer with a pH of 7.2. Simultaneously, the reaction medium in CSTR-2 with 24 h reaction was pumped into CSTR-3, which contained FSA$^{A129S}$ (30 U) in TEA buffer with a pH of 7.2. Finally, the reaction medium in CSTR-3 with 24 h reaction was pumped into a conical flask and collected for concentration and crystallization. The collected reaction solution was analyzed by high-performance liquid chromatography (HPLC). After reduced pressure distillation and vacuum drying, high-purity L-sorbose crystals were obtained.

## Reporting summary

Further information on research design is available in the Nature Portfolio Reporting Summary linked to this article.

## Data availability

The authors declare that all data supporting the findings of this study are available in the article and its Supplementary Information. Source data are provided with this paper.

## Code availability

All the tools used for the data analysis are publicly available, and the version and parameters used have been indicated. The custom pymol script used for the Ramachandran plot is available at Github: https://github.com/pymodproject/pymod/releases/download/v3.0/pymod3.zip. The qualitative model analysis of protein is available at: https://swissmodel.expasy.org/.

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

## Acknowledgements

This work was financially supported in part by National Key R&D Program of China (2020YFA0406103), NSFC (22232003, 21725102, 91961106, 91963108, 22175165), Strategic Priority Research Program of the CAS (XDPB14), Open Funding Project of National Key Laboratory of Human Factors Engineering (No. SYFD062010K), Youth Innovation Promotion Association CAS (2021451), USTC Research Funds of the Double First-Class Initiative (YD2060002025), Fundamental Research Funds for the Central Universities (WK2340000104, WK2400000004), and China Postdoctoral Science Foundation (2021M703063). The authors thank the support from USTC Center for Micro- and Nanoscale Research and Fabrication.

## Author contributions

G.L., Y.Z. and Z.L. contributed equally. C.G. and Y.X. supervised the project. G.L., Y.Z. and Z.L. designed and performed the experiments. G.W. designed the enzyme cascade reactors. F.G., C.Z., Y.W., H.Z., J.M., Y.H., A.C., J.P., Y.M., Z.T., C.G. and Y.X. discussed the data. G.L., Y.Z., Z.L., C.G. and Y.X. wrote the paper.

## Competing interests

The authors declare no competing interests.
