## [Peer Review File · Nature Communications]

Solar-driven sugar production directly from CO₂ via a customizable electrocatalytic-biocatalytic flow systemEditorial Note: This manuscript has been previously reviewed at another journal that is not operating a transparent peer review scheme. This document only contains reviewer comments and rebuttal letters for versions considered at *Nature Communications*. Mentions of the other journal have been redacted.

REVIEWER COMMENTS

[**Editorial note:** Reviewer #2 has assessed author's responses to Reviewer #1 and considers them adequately addressed.]

Reviewer #2 (Remarks to the Author):

Overall, the authors have made some significant improvements to the manuscript. Specifically, the decision to use an interlayered electrolyzer to reduce salt content has improved the merit of this work after demonstrating the negative impact of the salt in the downstream enzyme cascade. However, this work cannot be recommended for publication in *Nature Communications* in its current state given the severe lack of reporting surrounding the porous solid electrolyte interlayered electrolyzer. This lack of data makes the electrolysis step to produce the formic acid from CO₂ difficult for the reviewers to assess. Additionally, some further clarification on catalyst characterization and other more clarifications are still needed.

Given the authors' decision to switch to an interlayer design, the catalyst stability in this electrolyzer design needs to be reported. It is unclear if the post-reaction TEM images and XRD results were collected from catalysts used in the interlayer design or the traditional MEA design. The authors should clarify this. Additionally, it is common practice to report spent catalyst characterization following stability testing. The authors should provide XPS for oxidation state information and images of the catalysts before and after the 120 h stability test.

The reporting on the impact of salt concentration and the switch to an interlayer design to reduce effluent salt content to improve downstream biocompatibility is commendable.

However, the authors' reporting on the interlayer design and associated performance metrics are severely lacking which makes the electrolysis step of the proposed process difficult to evaluate. The following pieces of information need to be reported:

- Cell voltage during the stability test should be provided in the main text and current density should be reported rather than total current
- A justification for the likely increase in cell potential by switching to an interlayer design (ie. is the additional cell potential worth the reduced salt content to improve bioprocess performance?)
- A detailed schematic or photo of the flowfield design
- The interlayer thickness
- Electrode gasketing thickness along with the target compression for the Ti foam and carbon paper
- The resin preparation procedure
- The membrane preparation procedure
- Possible impurities the membrane (is the membrane exchanged to KOH form?)
- Possible impurities from the resin (the particles often come with trace impurities that need to be washed out)

Additionally, there appears to be a typo in the methods section where the Bi loading in the interlayered electrolyzer is mentioned, but a number is not provided.

The authors should add some brief discussion to the manuscript on the different pathways to produce formaldehyde and the advantages/disadvantages of each.

Reviewer #4 (Remarks to the Author):

To avoid bias, I read this paper first without referring to the three sets of reviews. My overall impression was that the paper contains a wealth of carefully prepared, detailed information that would be of interest to those in applied biotechnology. However, there is no new science and two issues make this unsuited for publication in Nature Communications. First, regarding novelty and notwithstanding the fact that efforts were made to improve enzyme properties, the work is effectively a well-designed optimization and application of several

'off-the-shelf' items rather than any new discovery: there are no interesting or important aspects attached to any individual item that would merit publication in this journal. Second, it is very difficult to see how such an elaborate hybrid system, employing a chemical reductant to regenerate NADH, is going to replace more conventional biotechnology or agriculture which is demonstrably much more scaleable. Without a complete lifecycle analysis, no strong case is made.

I then read the reviews.

Reviewer 1 seems primarily critical of the electrochemistry. Authors have addressed the points raised. They have also addressed concerns about formaldehyde reactivity and stability and the possible production of methanol. Reviewer 2 found the most interesting feature to be the production of a hexose starting from electrochemical reduction of CO₂, but is otherwise unsupportive, raising a number of technical points. Reviewer 3 describes the work as a variation of previous work then goes on to add further critical comments.

On the whole, none of these reviewers offers sufficient support for publishing this work in Nature Comm. As I state above, the manuscript is not without merit, but it conveys not a single new scientific principle. Although the experiments are carefully executed, the outcomes are nothing more than one would expect when joining together the many different components, in other words the science is no more than the sum of the parts.

Incidentally, I was just a little concerned about some elementary chemistry being lacking: on page 13, the authors describe their efforts to verify that BmfaldDH is thermodynamically capable of catalysing the formate-to-formaldehyde conversion. Why should they even question this fact? The enzyme, which does not have a redox cofactor, is a catalyst not a reagent.

Point-by-point response to the reviewers' comments

Reviewer #2:

Overall, the authors have made some significant improvements to the manuscript. Specifically, the decision to use an interlayered electrolyzer to reduce salt content has improved the merit of this work after demonstrating the negative impact of the salt in the downstream enzyme cascade. However, this work cannot be recommended for publication in Nature Communications in its current state given the severe lack of reporting surrounding the porous solid electrolyte interlayered electrolyzer. This lack of data makes the electrolysis step to produce the formic acid from CO₂ difficult for the reviewers to assess. Additionally, some further clarification on catalyst characterization and other more clarifications are still needed.

Author response: We really appreciate the referee's positive evaluation of our work, and are grateful to the referee for his/her comments and suggestions to help us further improve the quality of our manuscript. We have made all the revisions as suggested by the referee.

Given the authors' decision to switch to an interlayer design, the catalyst stability in this electrolyzer design needs to be reported. It is unclear if the post-reaction TEM images and XRD results were collected from catalysts used in the interlayer design or the traditional MEA design. The authors should clarify this. Additionally, it is common practice to report spent catalyst characterization following stability testing. The authors should provide XPS for oxidation state information and images of the catalysts before and after the 120 h stability test.

Author response: We thank the referee for his/her thoughtful suggestion. The catalyst stability in the porous solid-electrolyte reactor has been reported in Fig. 2e. The post-reaction TEM images and XRD results in Supplementary Fig. 10 were indeed collected from catalysts used in the solid-electrolyte reactor. According to the suggestion, we have clarified this point in the revised manuscript. Additionally, we have supplemented the relevant XPS data (Supplementary Figs. 4 and 11) and photographs (Supplementary Fig. 12) of the catalysts before and after the 120 h stability test and related discussion in the revised manuscript.

Supplementary Fig. 4 | Bi 4f XPS spectra of prepared Bi NWs before (a) and after (b) Ar ion etching.

Supplementary Fig. 11 | Bi 4f XPS spectra of Bi NWs after 120 h stability test in a porous solid-electrolyte reactor.

The Bi 4f XPS spectra of prepared Bi NWs (Supplementary Fig. 4a) could be divided into two groups according to binding energies: the two peaks at 158.9 and 164.2 eV are assigned to Bi^{3+} in Bi_2O_3 , while the other two peaks at 156.9 and 162.4 eV are assigned to metallic Bi^0 . The strong peak signal for Bi^{3+} and relatively weak peak signal of Bi^0 can be attributed to air oxidation of metallic Bi on the surface of Bi NWs. After Ar ion etching, the Bi 4f XPS spectra shows a much stronger signal of Bi^0 than that of Bi^{3+} (Supplementary Fig. 4b), confirming the inner metallic character of the prepared Bi NWs. The strong peak signal for Bi^{3+} after 120 h stability test (Supplementary Fig. 11) could be ascribed to the air oxidation of metallic Bi on the surface of Bi NWs after removing the negative potentials for CO_2 reduction reaction.

Supplementary Fig. 12 | Photograph of the Bi NWs catalysts coated on GDL. (a) before reaction; (b) after 120 h stability test (sticked with an anion-exchange membrane).

The reporting on the impact of salt concentration and the switch to an interlayer design to reduce effluent salt content to improve downstream biocompatibility is commendable. However, the authors' reporting on the interlayer design and associated performance metrics are severely lacking which makes the electrolysis step of the proposed process difficult to evaluate. The following pieces of information need to be reported:

Author response: We thank the referee for his/her suggestions to help us further improve the quality of our manuscript. We have provided all the information as suggested by the referee.

- Cell voltage during the stability test should be provided in the main text and current density should be reported rather than total current*

Author response: We thank the referee for his/her thoughtful suggestion. Cell voltage and current density during the stability test have been provided in the main text in the revised Fig. 2e.

- A justification for the likely increase in cell potential by switching to an interlayer design (ie. is the additional cell potential worth the reduced salt content to improve bioprocess performance?)*

Author response: We thank the referee for his/her thoughtful suggestion. In our original manuscript, the potential in the flow cell recorded in Fig. 2c is cathode potential rather than cell potential. Actually, the internal resistance in original system can reach up to 14 Ω and the cell voltage is 9.86 V at ~130 mA. According to the comments by Reviewer #1, we have employed a porous solid-electrolyte reactor to directly produce pure HCOOH solution, by which the internal resistance is reduced to 2.91 Ω . The real-time cell voltage of the solid-electrolyte reactor driven by the solar cell has been monitored (the revised Fig. 2e), indicating the cell voltage of our PV-EC system can keep stable in long-time test. The cell voltage of the solid-electrolyte reactor at the same ~130 mA current was reduced to 4.53 V. Thus, the cell potential has been significantly reduced by switching to an interlayer design.

- A detailed schematic or photo of the flowfield design*

Author response: We thank the referee for his/her thoughtful suggestion. The detailed photograph of the flow passage has been supplemented in the revised Supplementary Fig. 8.

Supplementary Fig. 8 | The digital image of the flow passage (a) and the solid-electrolyte reactor (b, c).

- *The interlayer thickness*

Author response: We thank the referee for his/her thoughtful suggestion. The thickness of the interlayered solid-electrolyte is 2.0 mm. We have provided this information in the Methods section for electrochemical measurements.

- *Electrode gasketing thickness along with the target compression for the Ti foam and carbon paper*

Author response: We thank the referee for his/her thoughtful suggestion. In a well-assembled solid-electrolyte reactor, the thicknesses of the electrode gasketing along with Ti foam and carbon paper are 1.0 mm (with a Ti foam thicknesses of 1.0 mm) and 0.30 mm (with a carbon paper thicknesses of 0.23 mm), respectively. We have provided this information in the Methods section for electrochemical measurements.

- *The resin preparation procedure*

Author response: We thank the referee for his/her thoughtful suggestion. The resin as solid-state electrolyte (AmberChrom 1 × 8 chloride form) was purchased from Sigma-Aldrich and was used as received without further processing. We have provided this information in the revised Supplementary Methods section.

- *The membrane preparation procedure*

Author response: We thank the referee for his/her thoughtful suggestion. The anion and cation exchange membranes were purchased from Dioxide Materials and Dupont, respectively. The membranes need further treatment before use. Specifically, the newly purchased anionic membrane (Sustainion X37-50 Grade 60) was soaked in a 1 M KOH solution for 8 hours. Then the membrane was continued to be activated for 24 hours in a new 1 M KOH solution to complete the activation process. After cutting to the appropriate size, the membrane was thoroughly washed with deionized water prior to being loaded in the solid-electrolyte reactor. The activated anionic membrane was stored in 1 M KOH solutions and the membrane was washed with deionized water several times prior to installing in the solid-electrolyte reactor.

The newly purchased cationic membrane (Nafion 117) was first soaked in a 5 wt% H₂O₂ solution at 80°C for 1 hour, followed by soaking in deionized water for 30 minutes. After that, the membrane was soaked in a 5 wt% H₂SO₄ solution at 80°C for 1 hour. Finally, the membrane was soaked in deionized water for 30 minutes to complete the activation process. The activated cationic membrane was stored in deionized water prior to installing in the solid-electrolyte reactor. We have provided this information in the revised Supplementary Methods section.

- *Possible impurities the membrane (is the membrane exchanged to KOH form?)*

Author response: We thank the referee for raising this concern. Just like the commonly applied procedure, the activated anionic membrane was stored in 1 M KOH solutions and the membrane was washed with deionized water several times prior to installing in the solid-electrolyte reactor. The anion and cation exchange membranes were purchased from Dioxide Materials (Sustainion X37-50 Grade 60) and Dupont (Nafion 117), respectively, and were further treated and washed before using. Thus, there should be no possible influential impurities. The ¹H NMR spectrum (Fig. 2d) for products over Bi NWs electrocatalyst in the solid-electrolyte reactor also confirms that there are no influential impurities. We have provided this information in the Supplementary Methods section.

- *Possible impurities from the resin (the particles often come with trace impurities that need to be washed out)*

Author response: We thank the referee for raising this concern. The resin as solid-state electrolyte (AmberChrom 1 × 8 chloride form) was purchased from Sigma-Aldrich. Normally, the porous solid-electrolyte reactor was first stabilized for 60 min before liquid product collection. During this process, the formed formic acid and possible trace impurities could be

washed out by the slow deionized water stream. Thus, there should be no possible influential impurities. The ^1H NMR spectrum (Fig. 2d) for products over Bi NWs electrocatalyst in the solid-electrolyte reactor also confirms that there are no influential impurities. We have provided this information in the Methods section for electrochemical measurements.

Additionally, there appears to be a typo in the methods section where the Bi loading in the interlayered electrolyzer is mentioned, but a number is not provided.

Author response: We thank the referee for his/her thoughtful suggestion. The Bi loading in the interlayered electrolyzer is 0.5 mg cm^{-2} . We have provided this information in the revised Methods section for electrochemical measurements.

The authors should add some brief discussion to the manuscript on the different pathways to produce formaldehyde and the advantages/disadvantages of each.

Author response: We thank the referee for his/her insightful suggestion. Direct conversion of CO_2 to formaldehyde under mild reaction conditions can be realized through the photochemical, electrochemical and enzymatic approaches (Angew. Chem. Int. Ed. 2022, 61, e202204008). To minimize overall energy consumption, the conversion of CO_2 to formaldehyde powered directly by renewable light or electricity would be preferable. However, the state-of-the-art photocatalysts and electrocatalysts for CO_2 conversion encounter the limitations with both low yield and selectivity for producing formaldehyde, which is prone to be reduced or oxidized. In contrast, enzymatic reduction of CO_2 displays high selectivity toward formaldehyde production. Nevertheless, the enzymatic reduction of CO_2 is normally a multi-enzymatic and multistep reaction, in which the related enzymes have modest activity and require cofactors as electron donors. To combine the complementary advantages of various approaches, we chose electrocatalytic CO_2 conversion, considering the substantially higher efficiency, to produce formate as the starting feedstock, which was coupled with one-step enzymatic catalysis for formate-to-formaldehyde transformation to enable the following enzyme cascade reactions. According to the suggestion, we have supplemented the related discussion in the revised manuscript.

Reviewer #4:

To avoid bias, I read this paper first without referring to the three sets of reviews. My overall impression was that the paper contains a wealth of carefully prepared, detailed information that would be of interest to those in applied biotechnology. However, there is no new science and two issues make this unsuited for publication in Nature Communications. First, regarding novelty and notwithstanding the fact that efforts were made to improve enzyme properties, the work is effectively a well-designed optimization and application of several ‘off-the-shelf’ items rather than any new discovery: there are no interesting or important aspects attached to any individual item that would merit publication in this journal.

Author response: We are grateful to the referee for his/her comments and would like to emphasize the significance of our work again. Although significant progress has been made on chemical conversion of CO₂ to C₁ and C₂ products through electrocatalysis or photocatalysis, it remains a great challenge to achieve sustainable synthesis of long-chain sugars by directly recycling CO₂. Despite the breakthroughs that have been made in previous reports by coupling two-step CO₂ electrolysis with microorganism fermentation (*Nat. Catal.*, 2022, 5, 388–396; *Nature Food*, 2022, 3, 461–471), the complex metabolic networks existing inside microbial cells pose a grand challenge to the customization of carbonaceous products. Moreover, knocking out the downstream transformation genes of microorganisms may cause serious negative impacts on sustainable sugar production. In addition, the commonly used model heterotrophic microorganisms require sugars as necessary nutrients, which cannot fully achieve carbon neutrality (*Nat. Catal.*, 2020, 3, 274–288; *Nat. Biotechnol.*, 2022, 40, 304–307). **In our work, for the first time, we report a hybrid electrocatalytic–biocatalytic flow system for achieving artificial synthesis of sugar directly from CO₂, by coupling photovoltaics-powered electrocatalysis with spatially separate enzyme cascade platform.** This enzyme cascade platform can be completely constructed *in vitro* by freely integrating different enzymatic reactions according to diverse upstream products from chemical CO₂ conversion, which will be more favorable for customizable food production in contrast to *in vivo* metabolic pathway and have no consumption of carbonaceous nutrients (*Science*, 2016, 354, 900-904; *Science*, 2021, 373, 1523-1527).

Second, it is very difficult to see how such an elaborate hybrid system, employing a chemical reductant to regenerate NADH, is going to replace more conventional biotechnology or agriculture which is demonstrably much more scaleable. Without a complete lifecycle analysis, no strong case is made.

Author response: We are grateful to the referee for his/her comments. We would like to point out that our designed system is not going to replace conventional biotechnology or agriculture. Instead, we aim to provide a new promising approach to supplement the traditional food production. The energy conversion efficiencies to biomass for most crop plants are only ~1% or less. More crucially, traditional food production by cultivation is facing the challenges from sustainable development, such as overuse of chemical pesticides and fertilizers, as well as encountering geographical restrictions, such as global climate change, land scarcity, and shortage of fresh water (*Nat. Sustain.* 2022, 5, 907-909). As such, new approaches to enhancing the

efficiency for food production, while reducing the dependency on natural resources and avoiding environmentally harmful chemicals, are greatly desired to supplement the traditional food production. Our approach to food production will particularly be available for applications in confined environments and physical space, such as space station, the region with atrocious circumstance or finite agricultural lands on earth. Additionally, we also adopted a reliable electrocatalytic method to regenerate NADH, thus completing this process without relying on chemical reductant (Supplementary Fig S18). Also, the recent work has reported that NADH can be efficiently regenerated by electrochemical method, which enables to avoid using phosphite in the future (*Nat. Commun.*, 2023, 14, 1814).

I then read the reviews.

Reviewer 1 seems primarily critical of the electrochemistry. Authors have addressed the points raised. They have also addressed concerns about formaldehyde reactivity and stability and the possible production of methanol. Reviewer 2 found the most interesting feature to be the production of a hexose starting from electrochemical reduction of CO₂, but is otherwise unsupportive, raising a number of technical points. Reviewer 3 describes the work as a variation of previous work then goes on to add further critical comments.

On the whole, none of these reviewers offers sufficient support for publishing this work in Nature Comm. As I state above, the manuscript is not without merit, but it conveys not a single new scientific principle. Although the experiments are carefully executed, the outcomes are nothing more than one would expect when joining together the many different components, in other words the science is no more than the sum of the parts.

Author response: We are grateful to the referee for his/her comments. We would like to point out that our manuscript was transferred from [journal name redacted] to *Nature Communications*, and we have made all the revisions as suggested by the referee. Reviewer #2 has assessed our responses to Reviewer #1 and considered them adequately addressed. Also, Reviewer #2 gave further comments and suggestions to help us further improve the quality of our manuscript.

Incidentally, I was just a little concerned about some elementary chemistry being lacking: on page 13, the authors describe their efforts to verify that BmfaldDH is thermodynamically capable of catalysing the formate-to-formaldehyde conversion. Why should they even question this fact? The enzyme, which does not have a redox cofactor, is a catalyst not a reagent.

Author response: We thank the referee for his/her insightful comments. We totally agree with the referee that there is no question on that BmfaldDH is thermodynamically capable of catalyzing the formate-to-formaldehyde conversion. We described our efforts to verify this because Reviewer #3 had proposed his/her concern about the formate-to-formaldehyde conversion by BmfaldDH.

REVIEWERS' COMMENTS

Reviewer #2 (Remarks to the Author):

The authors fully addressed my concerns. The quality of the manuscript is now suitable for a publication in Nature Communications.

Reviewer #2 (Remarks to the Author):

The authors fully addressed my concerns. The quality of the manuscript is now suitable for a publication in Nature Communications.

We thank the referee for his/her positive evaluation.